# Zero-Shot Adaptation of Behavioral Foundation Models to Unseen Dynamics

## Abstract

Behavioral Foundation Models (BFMs) proved successful in producing policies for arbitrary tasks in a zero-shot manner, requiring no test-time training or task-specific fine-tuning. Among the most promising BFMs are the ones that estimate the successor measure learned in an unsupervised way from task-agnostic offline data. However, these methods fail to react to changes in the dynamics, making them inefficient under partial observability or when the transition function changes. This hinders the applicability of BFMs in a real-world setting, e.g., in robotics, where the dynamics can unexpectedly change at test time. In this work, we demonstrate that Forward–Backward (FB) representation, one of the methods from the BFM family, cannot distinguish between distinct dynamics, leading to an interference among the latent directions, which parametrize different policies. To address this, we propose a FB model with a transformer-based belief estimator, which greatly facilitates zero-shot adaptation. We also show that partitioning the policy encoding space into dynamics-specific clusters, aligned with the context-embedding directions, yields additional gain in performance. These traits allow our method to respond to the dynamics observed during training and to generalize to unseen ones. Empirically, in the changing dynamics setting, our approach achieves up to a 2x higher zero-shot returns compared to the baselines for both discrete and continuous tasks.

## 1 Introduction

One very desirable property of reinforcement learning (RL) agents is their rapid adaptation to new tasks or to environment changes during test-time, without requiring any fine-tuning or planning. Achieving this in as few trials as possible would be even better: the ideal being the zero-shot adaptation [39], where the agent never interacts with the environment at test-time and relies solely on the data it was conditioned with. Behavioral Foundational Models (BFMs) [30, 37] may be considered as a step in this direction, because they can learn a variety of policies from offline data without knowing the rewards. During inference, it is possible to extract a task-specific policy that is optimal or near-optimal in terms of performance [38]. Recent work demonstrates [37] that one of the methods from the BFM family, based on Forward-Backward representation (FB) [38], is especially versatile and can successfully imitate behaviors from unlabeled data.

At the same time, FB possesses a fundamental drawback that limits its adaptation ability. In our paper, we show that FB is unable to generalize across different dynamics, such as changes in a transition function (*e.g.*, new obstacles) or an environment with some latent factor variation (*e.g.*, wind direction). This limitation stems from the way the *successor measure* [8] is estimated: FB averages the future-occupancy state distribution over all observed dynamics, which inevitably causes interference in policy representations. This fact alone may severely constrain the applicability of FB in the real-world scenarios. For example, one of the largest robotics dataset, Open X-Embodiment [7], consists of 22 different robot embodiments, and training FB on each of them simultaneously is infeasible. In Section 3.1, we discuss this limitation and support our claims theoretically.

Submitted to 39th Conference on Neural Information Processing Systems (NeurIPS 2025). Do not distribute.

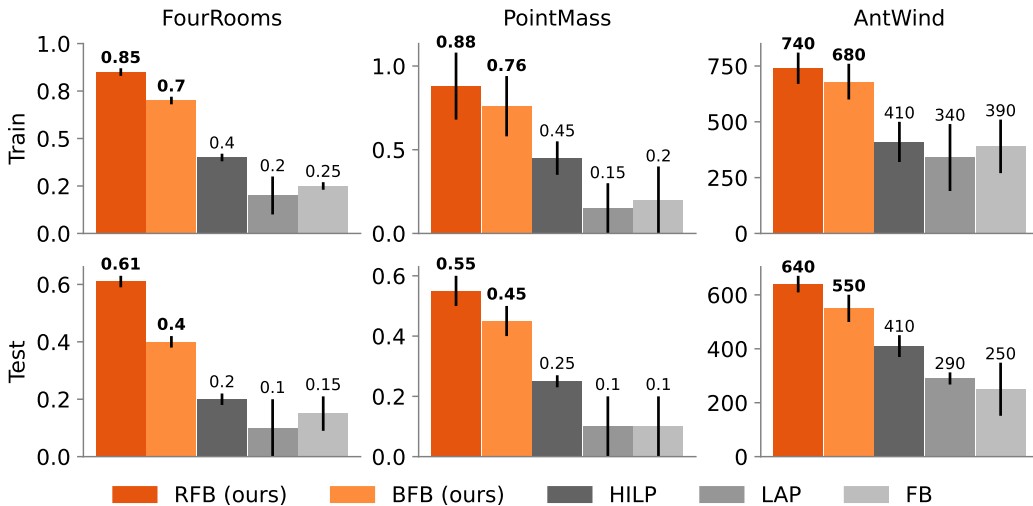

Figure 1: **Summary of results**. Aggregate mean performance over *seen* (train) and *unseen* (test) dynamics for zero-shot RL. The error bars indicate standard deviation over three seeds. Notably, both BFB and RFB adapt not only to the dynamics seen during training but are also able to generalize to unseen dynamics. There are 30 (20) training (test) dynamics for FourRooms and PointMass and 16 (4) for AntWind environments.

To remedy this, we introduce Belief-FB (BFB), a conditioning method for FB through a *belief* estimation, a popular technique of uncertainty quantification in Meta-RL [9, 46]. To implement this, we use a transformer encoder $f_{\text{dyn}}$ that, given a trajectory from data, outputs a dynamics-specific vector $h$ we then pass as a condition to the future outcomes representation function $F(\cdot, \cdot, h, \cdot)$. We pre-train $f_{\text{dyn}}$ in a self-supervised fashion, thus posing no additional requirements on the data structure or the trajectory re-labeling. We discuss the implementation of Belief-FB in Section 3.2.

Remarkably, Belief-FB enables the generalization capabilities of FB not only through the dynamics seen **in the training dataset**, but also on the **unseen test dynamics** never present in the offline data. We also find that in order to align *belief* estimation better with FB, one also needs to partition the policy space into dynamics-specific clusters, so we propose Rotation-FB (RFB) that accomplishes this partitioning. We present the theoretical support and the implementation details of Rotation-FB in Section 3.3. Empirically, both BFB and RFB outperform baselines for seen and unseen dynamics, as gathered in Figure 1 and discussed in Section 4.3.

We believe that our work sufficiently broadens the possible applicability of BFMs, yet keeping the zero-shot setting unchanged. Our contributions are as follows:

- **We introduce the limitation of Forward-Backward (FB) representations** [38], which lies in its inability to generalize *per se* across different dynamics both from train and test, where dynamics shift constitute of new layout grids or latent changes in the transition function that are hidden from an agent. Refer to Section 3.1 for more discussion.

- **We propose Belief–FB (BFB)**, which employs a transformer encoder to infer a belief over the agent's current dynamics [9, 46]. Analyzing BFB's policy space reveals that additional disentanglement is beneficial, motivating our Rotation–FB (RFB) extension. Section 3.2 examines Belief-FB, and Section 3.3 details Rotation-FB's theoretical motivation and implementation.

- **We empirically demonstrate that both BFB and RFB can adapt to different dynamics**, unlike its counterparts in the zero-shot setup. Refer to Section 4.3 for the discussion and Figure 1 for results.

## 2 Behavioral Foundation Models

For a reward-free Markov Decision Process (MDP), a Behavioral Foundation Model (BFM) [12, 27, 31, 37] is a RL agent trained in an unsupervised manner on a task-agnostic dataset of transitions. The

objective of a BFM is to approximate an optimal policy for a broad class of reward functions that are specified only at inference.

*Forward-Backward Representation (FB) [38]* approximates a successor measure for near-optimal policies across diverse tasks. The successor measure $M^\pi(s_0, a_0, X)$ for subset $X \subset \mathcal{S}$ is defined as cumulative discounted time spend at $X$ starting at $(s_0, a_0)$ and following $\pi$ thereafter. More formally, for tabular example:

$$M^\pi(s_0, a_0, X) = \sum_{t \geq 0} \gamma^t \mathrm{P}(s_t \in X | s_0, a_0, \pi), \tag{1}$$

with the corresponding Q-function for a specific task $r$:

$$Q_r^\pi(s_0, a_0) = \sum_{s^+ \in X} r(s^+) M^\pi(s_0, a_0, s^+). \tag{2}$$

In continuous case, the FB representation aims to approximate successor measure through finite-rank approximation under diverse policies through *forward* $F : \mathcal{S} \times \mathcal{A} \times \mathcal{Z} \to \mathbb{R}^d$ and *backward* $B : \mathcal{S} \to \mathbb{R}^d$ functions. Given a set of policies $\pi_z$ parametrized by task variable drawn uniformly from sphere $z_{\mathrm{FB}} \in \mathrm{Unif}(\mathcal{Z} = \mathbb{S}^{\lceil -\infty})$. Given $\rho$ as a probability distribution over states within the offline dataset, the objective for FB is written as:

$$M^{\pi_z}(s_0, a_0, X) \approx \int_{s^+ \in X} F(s_0, a_0, z)^T B(s^+) \rho(ds). \tag{3}$$

Then the policy can be obtained greedily as:

$$\pi_z(s) \approx \arg\max_a F(s, a, z)^T z. \tag{4}$$

For continuous case, the greedy policy is parametrized as Gaussian. During test time the task policy parametrization is approximated as $z_{test} \approx \mathbb{E}_{(s,a) \in \mathcal{D}_{test}} \{ r_{test}(s, a) B(s) \}$. If the inferred task vector $z_{test}$ lies within the task sampling distribution (in a linear span) $\mathcal{Z}$ used during training, then the optimal policy for task $r_{test}$ is obtained from Equation 2 as $\pi_z(s) \approx \arg\max_a Q_{r_{test}}^{\pi_z}(s, a)$. For more details on training and inference procedures of FB, we refer reader to Appendix A.3. More detailed discussion on the other related works is included in the Appendix A.

## 3  Method

**Problem Statement.** Our goal is to pre-train an agent in unsupervised regime in $\mathcal{C}_{train} = \{c_{train} \in \mathcal{C}\}$ contexts so that it is able to generalize to unseen ones during test time, *i.e.*, *zero-shot*[1]. We collect diverse dataset, consisting of mix of highly exploratory or expert-like unknown policies from varying environment layouts, differing either in dynamics (*e.g.*, wind, friction, etc.) or environment specifications (*e.g.*, positions of obstacles and doors). At test time, the agent is provided with small (up to episode termination steps) reward-free transitions from test context. Provided information must be used by an agent to recalibrate occupancy measure estimation corresponding to encountered environment. In an ideal scenario, the agent maximizes the expected discounted return across both train and test contexts. We refer to Appendix A for details.

To formally study optimality guarantees of the problem above, we employ the following assumption commonly used for dynamics generalization [10, 16]:

**Assumption 1** (Coverage). Let $\mathcal{P}^c(s_{t+1} | s_t, a_t)$ be a transition probability given small dataset of reward-free random interactions either from test or train context. Then, $\mathcal{P}^{c_{test}}(s_{t+1} | s_t, a_t) \Rightarrow \mathcal{P}^{c_{train}}(s_{t+1} | s_t, a_t) \; \forall s_t, s_{t+1} \in \mathcal{S}, a_t \in \mathcal{A}$.

### 3.1  Investigating latent directions space under multiple dynamics

We begin by addressing the following question: Why does FB representations fail to generalize effectively (both for train and test) to different situations under dynamics variations, *i.e.*, if learned on data sampled from diverse CMDPs? While the answer may appear intuitive, a closer look into the geometric structure of learned latent directions $z_{\mathrm{FB}} \in \mathcal{Z}$, which encode possible policies $\pi_z$ reveals critical insights which will be helpful later. We approach this question both theoretically and empirically on custom simplistic discrete partially-observable Randomized Doors (see Appendix

---

[1]We use the term "zero-shot RL" following [38].

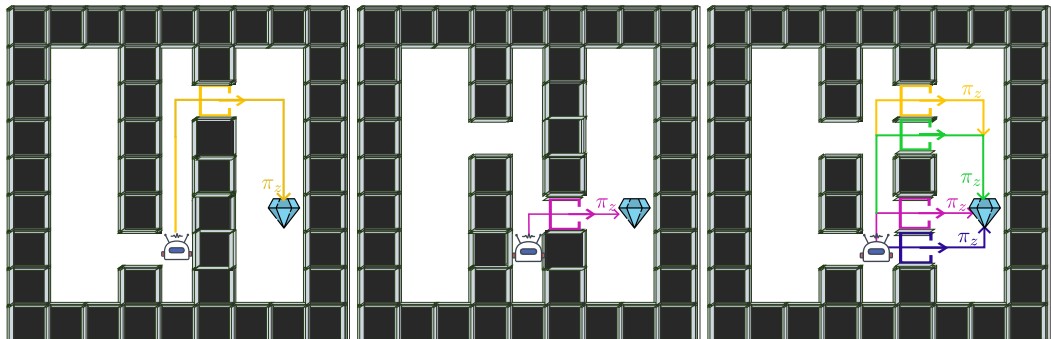

Figure 2: **Randomized-Doors environment for three different layouts, each produced through varying the grid structure (exact randomization procedure is a hidden variable)** (*left-middle*) From state $s$, the goal of an agent is to capture a diamond at target location by picking up the most probable policy $\pi_z$ (yellow for the first type and purple for the second) to move to the closest open door based on internal representation. (*middle*) When there are multiple possible future outcomes in the training data from the same state, the $\pi_z$'s (different colors) interfere with each other, leading to picking up an averaged policy.

B.1) environment. Partial observability adds additional challenges and showcases the need to estimate belief state, which we discuss in the following sections.

In this experiment the only source of dynamics variation is the grid layout type. That is, the positions of doors and walls are changed each new episode, depending on hidden configuration variable $c$. We collect a dataset of random trajectories drawn from multiple layouts, yielding near-uniform coverage of the entire $(x, y)$ states. Now, consider a particular state $s$ that an agent finds itself in three different layouts (see Figure 2). During FB training, we evaluate the forward representation $F(s, \cdot, z_{\text{FB}})$ for latent directions $z_{\text{FB}} \sim \text{Uniform}(\mathbb{S}^{d-1})$, where each $z_{\text{FB}}$ indexes a distinct policy starting at $s$.

In this setting a single grid state can require different optimal actions, depending on the layout an agent is instantiated in. Because $z_{\text{FB}}$ does not enforce a separation of layout-specific futures, the FB model suffers from *interference*: latent directions encoding conflicting future outcomes overlap and become entangled in the latent space $\mathcal{Z}$. For each of the layout configuration and fixed state $s$ from above, Figure 3 depicts latent directions $z_{\text{FB}}$, colored by optimal policy as $a_{\text{color}} = \arg\max_a F(s, a, z_{\text{FB}})^T z_{\text{FB}}$. When FB is trained on first two layouts in isolation, a unique dominant direction emerges in $\mathcal{Z}$, recovering the optimal goal-reaching policy $\pi_z^*$. In contrast, training on data which mixes transitions from various environment instances results in $z_{\text{FB}}$ to **blend dynamics-specific information** and instead to **average over the possible futures**, yielding a policy that is sub-optimal for every layout even from train set. Those observations are supported theoretically below.

Let $\{M^{\pi_i}\}_{i=1}^k$ be a collection of successor measures corresponding to optimal policies $\{\pi_i\}_{i=1}^k$ for distinct CMDPs defined by hidden context configurations $c_i \in C$. Assume that $\rho$ is the state-action distribution supported on the offline dataset used for FB training and $M^{\pi_i}(s, a, \cdot) \approx F(s, a, z_i)^T B(\cdot)$ is approximated via rank $d$ factors. Define the worst-case approximation error $\epsilon_k$ over context-dependent $k$ successor measures as follows:

$$\epsilon_k := \inf_{F, B} \max_{1 \leq i \leq k} ||M^{\pi_i} - F(\cdot, \cdot, z_i)^T B(\cdot)||_{L^2(\rho)}. \tag{5}$$

Then, the extracted policy $\pi_{z_i}$ for $(s, a)$ satisfies:

**Theorem 1** (Regret-bound for Multiple Dynamics). *For any bounded reward $||r||_\infty \leq R$ and particular test-time CMDP,*

$$\mathbb{E}_{(s,a) \sim \rho_{test}}[Q_r^{\pi^*}(s, a) - Q_r^{\pi_{z_i}}(s, a)] \leq \frac{2\gamma\epsilon_k ||r||_\infty}{(1-\gamma)^2}. \tag{6}$$

*Because $\epsilon_{k+1} \geq \epsilon_k$ (monotonicity), the worst case regret per any CMDP at test time increases as more environments are included during training.*

We provide a proof in Appendix. Intuitively, Theorem 1 tells that adding transitions from more CMDPs only increases the worst-case optimality gap: as number of environments $k$ grows, **FB is forced to average over incompatible future dynamics**. The proof relies on monotonicty property of

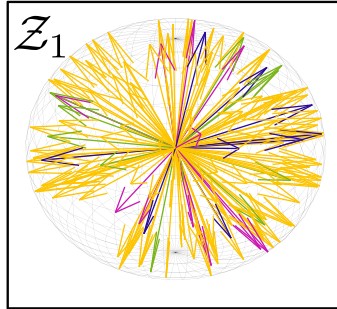 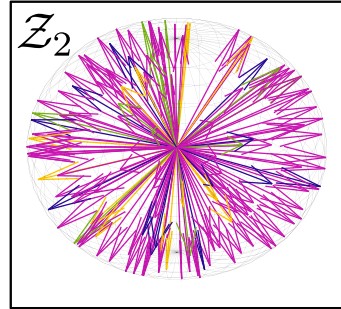 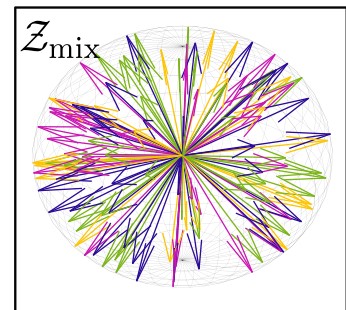

Figure 3: **Three different environment configurations from Figure 2 are visualized (yellow, purple and mixed trajectories).** For a fixed state $s$ and same goal across configurations, arrows depict latent directions $z_{\text{FB}} \in \mathcal{Z}$ and colored by optimal action as $a_{color} = \arg\max_a F(s, a, z_{\text{FB}})^T z_{\text{FB}}$. (*left-middle*) When FB is trained on the two distinct configurations in separation, most of the latent directions agree on the optimal policy $\pi_z$. (*right*) When FB is trained on mix of CMDPs and at test time tasked with any particular configuration from train, obtained policy is ambiguous, since most policy-encoding directions do not agree on the action.

141  error term in Equation 5 and Theorem 9 from Touati and Ollivier [38]. In Section 3.3 we will refine
142  this result and show that it is possible to remove explicit dependence of $k$, lowering the upper bound.

143  This interference highlights a fundamental trade-off. FB is expressive enough to model any task, yet
144  when it is trained in unsupervised manner across environments with distinct unobserved parameters,
145  the lack of contextual conditioning forces it to average different dynamics rather than separate them.
146  The resulting successor measure merges transitions from distinct layouts and entangles directions in
147  the latent space $\mathcal{Z}$. To disentangle these directions we must represent uncertainty about the hidden
148  context explicitly. The next section introduces a belief-conditioned objective that infers the latent
149  context and allows FB to maintain environment-specific successor features.

> **Takeaway 1**
>
> Because FB training inherently averages over all possible future states, it cannot learn a
> disentangled policy space and, therefore, fails to adapt to changes in dynamics.

150

## 3.2   Belief State Modeling

152  To resolve the interference issue described in Section 3.1, we **infer the latent context of an envi-**
153  **ronment and augment FB input on that belief**. We train a transformer encoder $f_{\text{dyn}}$, by taking a
154  *set* of transitions $\{(s_t, a_t, s'_{t+1})\}_{t=1}^N$ and outputs an embedding $h \in \mathbb{R}^d$. We denote the space of all
155  possible inferred contexts as $\mathcal{H}$, where each element $h$ encodes dynamics for particular environment.
156  Because the ordering is discarded and no rewards in transitions are provided, the encoder must focus
157  on dynamics specific mismatches (*e.g.*, layout geometry, friction or wind direction), rather than
158  policy specifics. Such context encoder should be permutation invariant, since unobservable factors
159  describing environment are independent of the order of transitions in an episode. This setting provides
160  theoretical ground for zero-shot and few-shot learning [33].

161  Concretely, dataset consists of episodes $(\{(s_t, a_t, s'_{t+1})_{c_i}\}_{t=1}^N$ coming from CMDP with randomly
162  instantiated hidden specification variable $c_i$ (different dynamics). We train a transformer encoder on
163  random episodes (without episodic labels $c_i$) of context length $n$ to infer contextual (hidden) variable
164  $h$ which fully specifies the dynamics across given episode. The transformer encoder loss involves two
165  main components: 1) $h$ is encouraged to follow a Gaussian prior and is shared across trajectory, and
166  2) projection head, which combines $h$ with $(s_t, a_t)$ to predict $s_{t+1}$. Those stages can be either trained
167  end-to-end or separately. We observed that separating FB training from $f_{\text{dyn}}$ gives better results.

168  For each trajectory we concatenate the inferred context vector $h$ with the task vector $z_{\text{FB}}$ to obtain
169  augmented input $[h; z_{\text{FB}}]$ and condition only forward network as:

$$\hat{M}_{\pi_z}(s_t, a_t, s_{t+1}) = F(s_t, a_t, [h; z_{\text{FB}}])^T B(s_{t+1}). \tag{7}$$

170  We empirically found that conditioning the backward network $B$ degraded performance, producing
171  smoothed out $Q$ function, ignoring environment structure, so in our experiments $B$ remains shared
172  across contexts. Training procedure is summarized in Algorithm 1.

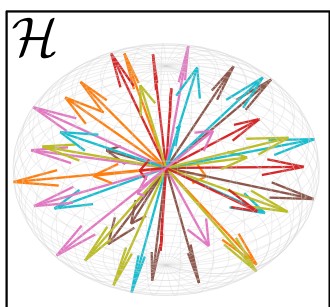 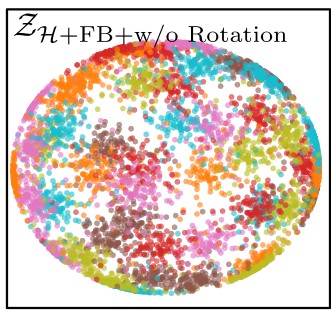 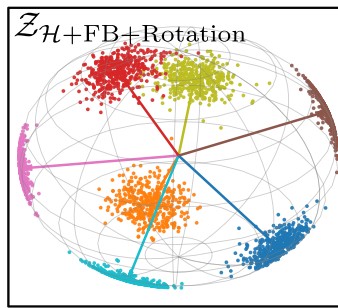

Figure 4: **Visualization of inferred contexts $h$ from space of all possible contexts $\mathcal{H}$ (depicted as arrows) and task vectors $z_{\text{FB}}$ (depicted as points on sphere boundary). Transitions from same CMDP colored the same. Concentration parameter $\kappa$ defines spread of clusters.** (*left*) Untrained transformer $f_{\text{dyn}}$ output for different transitions is unstructured and same transitions coming from same CMDP (identical colors) are not collinear. (*middle*) New sampling procedure aligns policy specific vectors $z_{\text{FB}}$ with context specific $h$, but clusters overlap before training. (*right*) After training, $h$ for transitions from the same context are aligned and policies $z_{\text{FB}}$ do not interfere between different environment configurations.

At test time, the agent is provided with a short, reward-free trajectory and it is passed to $f_{\text{dyn}}$ to obtain $h$. By plugging the result into Equation 4, the greedy policy is obtained.

> **Takeaway 2**
>
> We train a transformer in a self-supervised regime to estimate a belief over possible contexts, augmenting FB inputs and enabling effective disentanglement of contextual representations.

## 3.3 Structuring directions in the latent space

Insights from Section 3.1 showed that sampling task-vectors $z_{\text{FB}}$ uniformly on the hypersphere encodes averaged policies, while Section 3.3 provided a solution through explicit context identification. We now combine these observations together through enhanced sampling $z_{\text{FB}}$ around the inferred context $h$.

In Vanilla-FB, each state $s$ draws $z_{\text{FB}} \sim \text{Unif}(\mathbb{S}^{d-1})$ with no inductive bias, so resulting policies $\pi_z$ conflict with each other in CMDP setting, **even if additional explicit conditioning is introduced as before**. We replace uniform prior with a *von Mises-Fisher*(vMP) distribution centered at the context direction for episode $h = f_{\text{dyn}}(\{(s_i, a_i, s_{i+1})\})$ as

$$z_{h+\text{FB}} \sim \text{vMF}(\mu = h, \kappa). \tag{8}$$

with $\kappa$ controlling the spread or *diversity* of policies (left and middle figures from Figure 4). In practice, to draw $z_{h+\text{FB}}$ we first pick a simple vector (*e.g.*, the first basis vector), perturb with vMF noise, and finally rotate the result onto $h$ with Householder reflection.

This enhancement has several benefits: 1) because directions $h$ that differ in dynamics now occupy disjoint cones on the hypersphere, FB can fit the successor measure locally inside each cone, avoiding the destructive averaging effect quantified in Section 3.1 and 2) alignment procedure encourages the agent to explore policies that are plausible under its current belief while still injecting controlled diversity through $\kappa$.

Importantly, such a procedure not only has empirical benefits as we will show in Section 4, but also lowers bound from above in Theorem 1, *making it non dependent on number of environments $k$.*

**Theorem 2** (Regret bound under latent space partitioning). *Define $\epsilon_k$ as worst-case approximation error as in Equation 5. The Gram matrix of the task directions $\{z_{FB}\}_{i=1}^{k}$ is block diagonal w.r.t. partition $\{S_j\}$, with each $S_j$ being the set of task-vector indices which satisfy $\langle z_{FB}, h^j \rangle \geq \cos\theta_{max}$ with $\theta_{max}$ being angle between any two latent vectors. Then,*

$$\epsilon_k = \max_{j \leq L} \epsilon_j, \quad \epsilon_k \leq \epsilon_{k_{max}}, \tag{9}$$

*with $k_{max} := \max_j |S_j|$ being the size of the largest cone block.*

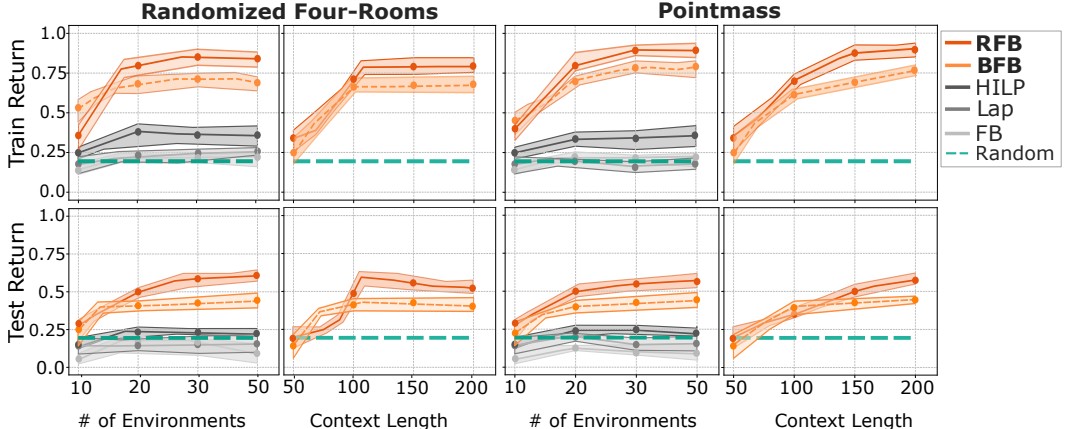

Figure 5: **Ablations on data diversity and context length of transformer encoder.** We show the influence of number of environments (data diversity) and context length on train and test performance in Four-Rooms and Pointmass environments. For data-diversity ablation, we see a clear performance boost up until some point, after which it plateoes, as the Theorem 1 predicts. In our context-length ablation, we observe similar behaviour: performance improves as the context grows up to the length of a single episode, and then levels off. The results are averaged across three seeds, the opaque fill indicates standard deviation.

Intuitively, Theorem 2 states that after the partitioning procedure of the latent space into non-overlapping clusters based on context representations $h$, the global worst-case FB approximation error $\epsilon_k = \max_{j \leq L} \epsilon_j$ is determined only by the cluster whose error $\epsilon_j$ is largest. Importantly, this bound *does not depend on number of training environments $k$*. We provide a more formal treatment and a full proof in Appendix D.

---

**Takeaway 3**

Adjusting the prior over task vectors $z_{\text{FB}}$ further mitigates the averaging effect and disentangles policy representations better.

---

## 4 Experiments

In this section, we compare proposed methods, namely: **Belief-FB (BFB)** (Section 3.2) and its enhancement **Rotation-FB (RFB)** (Section 3.3), against the baselines in both discrete and continuous settings. We outline each experiment design below; all necessary details are provided in Appendix C. Every environment is framed as a contextual MDP (CMDP), where the context differs by the underlying hidden variation (*e.g.*, , grid layout or transition dynamics). During test time, we provide a single trajectory from random policy, which enables context configuration inference.

### 4.1 Environments and Setup

To support claims and theoretical insights made in previous sections, we consider the following experimental setups: **(i)** discrete, partially observable Randomized Four-Rooms (Appendix B.2), **(ii)** continuous AntWind (Appendix B.3), and lastly **(iii)** continuous partially observable Randomized-Pointmass (Appendix B.4). We vary the number of train layouts for each experiment, while fixing the number of held-out *unseen* context settings to 20 for Randomized Four-Rooms and Randomized-Pointmass, and 4 for Ant-Wind. We perform comparisons against following baselines:

**HILP** [26] is a method that learns state representations from offline data so that the distance in the learned representation space is proportional to the number of steps between two states in original space. **FB** [38] is an original version of the FB, described in Section 2. **Laplacian RL (LAP)** [42] constructs a graph Laplacian over state transitions from experience replay, then computes its eigenvectors to form low-dimensional representations that capture the environment's intrinsic structure. **Random** agent, which randomly explores the environment in a task-independent manner.

**Randomized Four-Rooms** is a discrete, deterministic, partially observable environment, where the task is to optimally move to the goal location. Training data is collected by executing random policies in $N$ distinct grid layouts, that differ in doorway and wall locations.

**Ant-Wind** is a continuous environment, where the goal is to make a four-legged ant walk forward as fast as possible. The environment dynamics are determined by the direction (angle) of a wind $d$.

**Randomized-Pointmass** is a partially observable continuous environment, where the task is to move to the goal locations. Maze grid structure is generated randomly, where each cell either contains wall or empty, while ensuring there is a path between start and goal locations.

### 4.2 Can the belief estimation enable adaptation in FB?

Previously, we provided the theoretical foundations and speculated on the matter why FB is unable to differentiate between distinct dynamics and how we can use the belief estimation to overcome this. We refer to Table 1 and Figure 1 that show our empirical findings to support our claims.

Initially, we would like to highlight that neither FB nor LAP are able to outperform a simple random baseline in PointMass and FourRoom, indicating that the policy they learn is most likely stuck in some obstacle due to averaging (see Section 3.1. Only HILP, which uses a different way to learn policy representations, is able to perform better than random policy.

In contrast, Belief-FB and Rotation-FB outperform every baseline method, indicating that belief estimation is indeed a missing piece for adaptation. Notably, our methods also demonstrate generalization capabilities beyond train data on unseen test tasks.

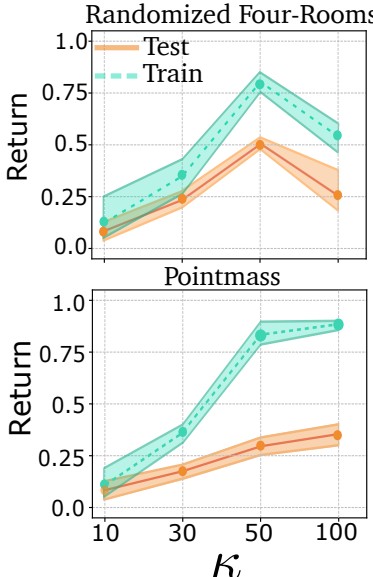

Figure 6: Influence of $\kappa$ in RFB on performance. The results are averaged across three seed, the opaque fill represents standard deviation.

### 4.3 Do BFB and RFB capture hidden properties of the environment?

For an agent to refine its policy, it needs to keep track and update the uncertainty over possible environment configurations. Both Belief-FB and Rotation-FB accomplish this. Figure 7 illustrates this phenomenon visually. In Randomized-Door (left), the episodic trajectories from five layouts form non-overlapping clusters in the first two principal components of $h$, effectively disentangling different dynamics.

In Ant-Wind, the embeddings lie almost perfectly on a circle whose azimuth matches the underlying wind direction, generalizing smoothly to the 4 held-out wind angles. The quantitative results for evaluation in Table 1 (averaged across all environments) reveal that the baseline methods fail to recover those environment-specific properties and therefore produce sub-optimal policies even for train cases. In particular, HILP tends to predict an average direction in Randomized Four-rooms and ignores obstacles, while FB outputs same policy and $Q$ function for almost all environments. Figure 12 shows that $Q$ function is properly estimated only for BFB and RFB, respecting wall positions.

### 4.4 Does change in context length input to the $f_{dyn}$ impacts performance?

In this experiment, we examine whether increasing the input trajectory length of improves performance. We vary the context length of $f_{dyn}$ from 50 to 200 and present the results in Figure 5 for both Randomized Four-Rooms and Randomized Pointmass environments, across train and test configurations. The results show that performance is poor when the context length is shorter than a single trajectory episode (100 steps), as short trajectories only capture local, near-term goals. Conversely, excessively long sequences provide no additional benefit due to redundancy, since $f_{dyn}$ already contains all neccessary information. Evaluations on both train and test environments demonstrate that $f_{dyn}$ produces representations $h$ capable of distinguishing between different context instances while maintaining robustness.

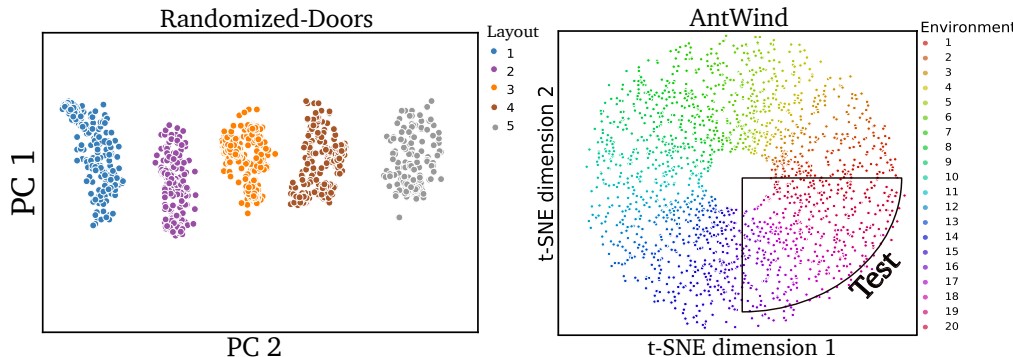

Figure 7: **2D projections of $z_{\text{dyn}}$ inferred from different trajectories across number of different contexts (color), showing effective disentangling environments based on transition function or other mismatches.** (*left*) First two principal components are visualized for estimated $z_{\text{dyn}}$ from five trajectories, each representing different layout type in Randomized-Doors. (*right*) Inferred context variables $z_{\text{dyn}}$ recover hidden wind direction parameter in AntWind environment both for train and test, proving successful extrapolation properties.

### 4.5 Does increase in dataset diversity make policies more robust?

We study whether diversifying training configurations of CMDPs results in better performance. Intuitively, larger the state-action space coverage, successor measure estimation should be more accurate. This intuition is also reflected in experiments: Figure 5 depicts that up to some number $N$ (around 25) improvement rapidly grows for BFB and BFB, while baselines perform on par with random policy, supporting insights from previous sections. Once learned representations $h$ from $f_{\text{dyn}}$ covered all modes of variation (*i.e.*, contexts), adding more data yields marginal benefit ($< 3\%$) marginal gain. These findings align with theoretical intuition from Theorem 1.

### 4.6 How $\kappa$ in RFB influences performance?

As described in Section 3.3, RFB concentration $\kappa$ regularizes the diversity of policies for each environment. One the one hand, concentration should be high to ensure non-overlapping policy parametrized clusters $\pi_z$ for different $h$, while at the same time it should not exceed certain value to control the diversity of policies in the environment, preventing collapsed solutions. Figure 6 shows that lower values of $\kappa$, meaning task-vectors $z_{\text{FB}}$ are sampled with high deviation around $h$, likely producing overlapping clusters. As $\kappa$ grows, task-vectors become more specialized, lowering variance which results in higher performance.

## 5 Conclusion & Limitations

In this work, we introduce **Belief-FB (BFB)** and **Rotation-FB (RFB)** two methods that extend the Forward-Backward (FB) representation to handle novel dynamics. We first identify a critical limitation in existing approaches: interference arises when naively sampling policy-parametrized latent directions during training on transitions from conflicting dynamics. To address this, we learn hidden context variables (belief states) via a permutation-invariant transformer encoder and use them for additional conditioning. We further improve latent-direction sampling by aligning task-relevant abstractions with environment-specific features, ensuring non-overlapping regions in latent space of policies. Both BFB and RFB demonstrate theoretical and empirical improvements over prior methods. However, limitations include evaluations on a narrow set of dynamics mismatches and the introduction of the additional hyperparameter $\kappa$ that controls policy diversity across environments. Also, usage of transformer can be expensive if context length grows.

As future research directions, it would be valuable to investigate whether other zero-shot RL methods, those not based on successor-measure estimation, exhibit similar interference issues, and to scale our approach to more complex benchmarks such as XLand-MiniGrid [24, 25] or Kinetix [22].

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

# A Extended Related Works and Background

## A.1 Background

**Contexual Markov Decision Process.** Throughout paper we will be dealing with a Contextual Markov Decision Process (CMDP), defined by a tuple $\langle \mathcal{C}, \mathcal{S}, \mathcal{A}, \gamma, \mathcal{M} \rangle$, where $\mathcal{C}$ is a context space and $\mathcal{S}, \mathcal{A}$ are shared state and action spaces across environments. Function $\mathcal{M}$ maps particular context $c \in \mathcal{C}$ to respective MDP, *i.e.*, $\mathcal{M}(c) = \langle \mathcal{S}, \mathcal{A}, \mathcal{T}^c, R^c, \mu^c, \gamma \rangle$ with context-dependent transition function $\mathcal{T}^c : \mathcal{S} \times \mathcal{A} \times \mathcal{C} \to \mathcal{S}$, $\mu^c$ being an initial distribution over states and $\gamma \in (0,1)$ a discount factor. Intuitively, the context $c \in \mathcal{C}$ represents a fixed environmental configuration, such as obstacle positions, layout geometry, dynamics vector parameters or seed. Throughout this work, the context remains static within each episode, consistent with prior literature [18, 23, 36]. A policy $\pi : \mathcal{S} \to \Delta\mathcal{A}$ is optimal for context $c$ for the reward function $R$ if it maximizes expected discounted future reward, *i.e.*, $\pi^*_{c,R}(s_0, a_0) = \arg\max_\pi \mathbb{E}[\sum \gamma^t R(s_t, a_t)|s_0, a_0, \pi, c]$.

When the context is fully observable, augmenting the state space with the given context reduces the CMDP to a standard MDP, eliminating the need to model distinct dynamics $\mathcal{T}^c$, rewards $R^c$ or initial states $\mu^c$. However, if the context is partially observable, the learned model must infer and track the uncertainty over true hidden configuration to maintain theoretical optimality guarantees. Such task can be framed as posterior estimation $p(c|\mathcal{H})$ or *belief* over possible contexts $c$ given accumulated history $H$.

Most successful methods for deriving an optimal policy across arbitrary tasks from a task-agnostic dataset leverage successor features [2, 6, 8, 26, 45] or their continuous counterpart, successor measures [1, 5, 17, 38, 39]. In this work, we focus on the latter framework, specifically its instantiation via forward-backward representations [38]. Below, we briefly outline its key properties.

**Zero-Shot RL.** Given an offline dataset of transitions $\mathcal{D} = \{(s_i, a_i, s_{i+1})\}_{i=1}^{|\mathcal{D}|}$ generated by an unknown behavior policies, the agent's objective is to learn a unified abstraction of the environment without additional interaction. At test time, this abstraction helps to obtain optimal policy for *any* reward function $r_{test}$ which defines a particular *task*. Reward function can be specified either as a small dataset of reward-labeled states $\mathcal{D}_{test} = \{(s_i, r_{test}(s_i)\}_{i=1}^k$ or as a direct mapping $s \to r_{test}(s)$. While some prior works assume access to the context labels [14], we focus on the setting where the context is unknown and must be inferred from the data. Alternative formulations of zero-shot RL exist under other formalisms, and we refer to [18] for comprehensive overview.

## A.2 Literature

**Domain Adaptation and Transfer Learning in RL.** While our work will focus on domain adaptation applied to estimating successor measure for various dynamics mismatches, we start by briefly reviewing more general ideas in classic domain adaptation and refer to [19] for detailed overview. Most methods for domain adaptation can be categorized into *importance-weighting* [4, 34, 40] and *domain-invariant feature learning* [10, 11, 43, 44] approaches. Former methods estimate the likelihood ratio of examples under samples from target domain versus samples from source, which is then used to recalibrate examples from the source domain. The latter approaches learn a unified representation of the environment, targeting to extract only task-relevant abstraction, negating distracting information.

The most relevant approach which enables FB representations to generalize across dynamics is *Contexual FB* [16]. This approach uses importance-weighting formalism and introduces two classifiers, which estimate the likelihood of transitions $(s_t, a_t)$ and $(s_t, a_t, s_{t+1})$ being from train or test context and augment the reward function to account for those discrepancies in the dynamics. If augmented reward function lies in the linear span of the $\mathcal{Z}$ space during FB training, then the policy can be extracted as described in Equation 4. However, such an approach requires training classifiers from scratch for each novel layout of the environment, limiting its applicability.

**Meta-RL.** Another major line of related works, Meta-Reinforcement Learning (Meta-RL), focuses on few-shot domain adaptation to unseen tasks or dynamics [3]. The significant part of research in Meta-RL is dedicated to explicitly learning the *belief* by collecting a history of interactions with the environment on inference during test-time [9, 29, 46]. However, recent works show that it is possible to quantify the *belief* without learning the posterior implicitly [20, 21, 28, 32, 35, 47, 48]. Leveraging in-context ability of transformers [41], one can learn an end-to-end supervised model, while the transformer's context will absorb into robust representation the adaptation-relevant information thus

835 enabling fast adaptation. We also leverage this in-context ability to construct the belief representation
836 of the dynamics the agent currently in, but instead operating in a zero-shot manner.

## A.3 FB Training

838 In this section we describe the training procedure of FB in more details. Everything follows the
839 notation from Touati and Ollivier [38].

840 Assume that $\rho$ is supported over all provided data, *i.e.*, it is non-zero everywhere.

$$\mathcal{L}_{\text{FB}} = \mathbb{E}_{(s_t, a_t, s_{t+1}, s_+) \sim \mathcal{D}, z \sim \mathcal{Z}}[(F(s_t, a_t, z)^T B(s_+) - \gamma \hat{F}(s_{t+1}, \pi_z(s_{t+1}, z)^T \hat{B}(s_+))^2 \\ - 2F(s_t, a_t, z)^T B(s_{t+1})] \quad (10)$$

841 Here, $s_+$ is a future outcome either from the same trajectory or randomly sampled from data.
842 $\hat{F}, \hat{B}$ are target networks with $Z$ being a task space, encoding all possible policies. The policy
843 $\pi_z$ is trained in an actor-critic formulation and parametrized as Boltzmann policy $\pi_{z_i}(\cdot|s_i) = $
844 $\texttt{softmax}(F(s_i, \cdot, z_i)^T z_i / \tau)$ for continuous environments. Additionally, $B$ is forced to be orthogonal
845 for different s, which is enforced by contrastive loss $\mathbb{E}_{(s, s_+)}[B(s)^T B(s_+)]$.

# B  Environment Descriptions

## B.1  Randomized-Doors

848 The Randomized-Doors MiniGrid environment (Figure 8) is a discrete-state, discrete-action finite
849 horizon deterministic environment in which agent has an objective to go to goal location with
850 maximum return of $1$. Each episode terminates after 100 steps or after reaching goal location. The
851 randomization determines possible open doors locations, fully specifying particular layout. In our
852 experiments, the observation state of an agent consists of $(x, y)$ coordinates tuple, making it partially
853 observable. Such setting requires to properly update beliefs over unobservable layout configuration
854 type. The action space consists of four actions, namely {up, down, right, left}, while $(x, y)$
855 coordinates across both axes are bounded by grid size, which we take to be $9 \times 9$.

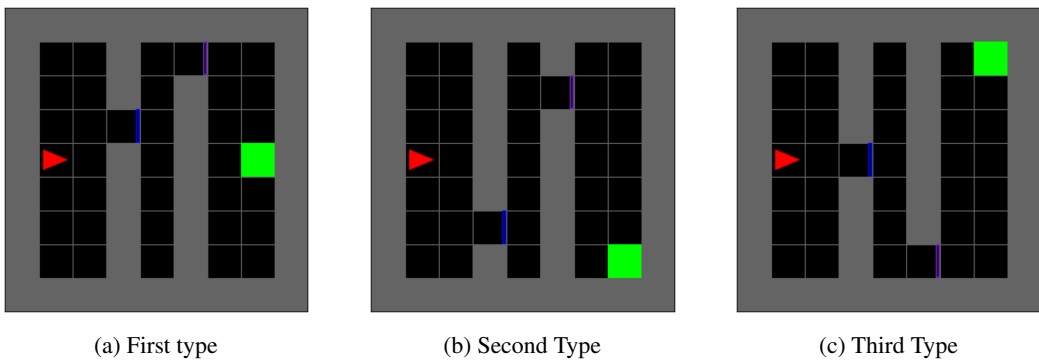

(a) First type       (b) Second Type       (c) Third Type

Figure 8: Several possible layouts are visualized, each corresponding to unique possible doors
configurations. The agent is denoted as a red triangle. The task specification (goal position) with
reward of 1 is denoted by green square and is also randomized. It is a custom implementation based
on Empty MiniGrid (https://minigrid.farama.org).

## B.2  Randomized Four-Rooms

857 The Randomized Four-Rooms MiniGrid environment Figure 10 is a modification of classic Four-
858 Rooms and is a discrete-state, discrete-action, deterministic partially observable environment. For
859 each episode, the maze layout (grid type) is generated randomly, ensuring all of the four rooms are
860 connected with exactly single door. Observation state consists of $(x, y)$ coordinates, making this
861 environment hard and checks whether agent could successfully estimate uncertainty over hidden
862 configurations solely based on number of occurrence of each transition, recovering dynamics. In our
863 experiments, we consider $11 \times 11$ bounds for height and width.

864 Observation space consists of raw discrete $(x, y)$ coordinates on the grid, while actions correspond
865 to a set of possible moves {up, down, left, right}. For every layout we record $500$ episodes

of length 100, yielding a dataset that covers almost all possible $(s, a)$ transitions. For testing on unseen configurations, we fix agent starting position to coordinates of the first empty cell and evaluate performance across 3 static goal positions, farhest away from starting position.

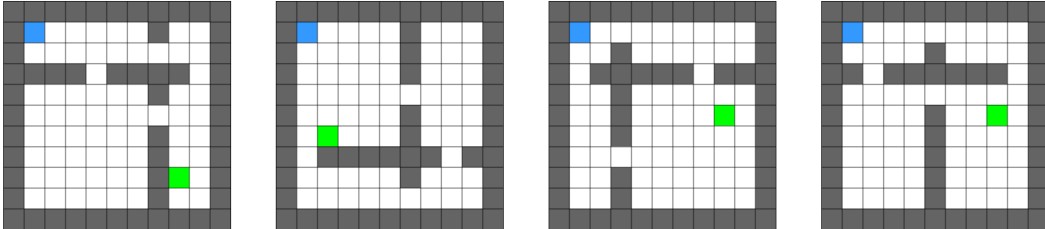

Figure 9: **Different layout configurations from randomized Four-Rooms environment.** During inference, the goal for the agent (depicted in blue) is to achieve green location. In our experiments we fix starting agent position and fix 3 goals, one for each room.

### B.3 Ant-Wind

The AntWind environment is a modified version of the Ant locomotion task from the MuJoCo simulator, commonly used to test an agent's adaptability to changing dynamics. In this environment, an ant-like robot must learn to move forward while being subjected to external wind forces varying in magnitude and direction. In our experiments we consider 17 environments for training, covering three quadrants of possible wind directions on the circle, while leaving others for test, checking extrapolation on the fourth quadrant.

For our experiment, we collect dataset by training SAC [15] on $3/4$ of all possible directions, which results in 16 environments and hold out the other $1/4$ for evaluation. Resulting dataset consists of 3400 transition tuples, where each environment configuration is represented as trajectory of length 256.

### B.4 Randomized Pointmass

Randomized Pointmass is a modification of pointmass environment from D4RL [13]. Each episode the environment grid structure is randomized, ensuring all cells are interconnected. The observation space consists of $(x, y)$ transitions. Start position is determined as a first empty cell, while goal location is chosen to be the fartherst away from start (based on Manhattan distance) and ensuring existence of at least one valid trajectory (*e.g.*, through BFS).

Observation space consists of (global $x$, global $y$) position, similar to Four-Rooms. We fix dataset size to be $1e^6$, only varying number of layouts and episodes per layout, while fixing episode length to 250. We use explore policy, which is a random policy with a portion of actions repeated ("sticky-actions").

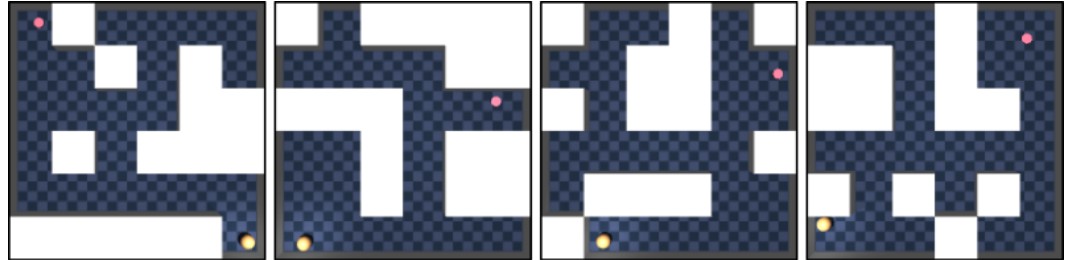

Figure 10: **Examples of pointmass grid variations.**

## C Experiments Details

**Randomized-Doors.** For didactic example from Section 3.1 we collect diverse dataset from different layout configurations (open door locations) such that visitation distribution over all states is non-zero. Black color denotes obstacles. The episode length is set to be 100, which is equal to the context

length of the transformer encoder for this experiment. Overall, we collect $500$ episodes per layout and coverage heatmap is visualized in Figure 11.

Table 1: Comparison of proposed approaches against baselines on **test** (unseen) environments. Results for Fourrooms and Pointmass are averaged across 20 mazes configurations.

| Environment (Test) | Method | | | | | |
|---|---|---|---|---|---|---|
| | Random | Vanilla-FB | HILP | Lap | **Belief-FB** | **Rotation-FB** |
| Randomized-Fourrooms | $0.05_{\pm 0.01}$ | $0.15_{\pm 0.06}$ | $0.2_{\pm 0.02}$ | $0.1_{\pm 0.1}$ | $0.4_{\pm 0.02}$ | $0.61_{\pm 0.02}$ |
| Randomized-Pointmass | $0.03_{\pm 0.01}$ | $0.1_{\pm 0.1}$ | $0.25_{\pm 0.02}$ | $0.1_{\pm 0.1}$ | $0.45_{\pm 0.05}$ | $0.55_{\pm 0.05}$ |
| Ant-Wind | $250_{\pm 200.0}$ | $250_{\pm 98.5}$ | $410_{\pm 40.5}$ | $290_{\pm 22.5}$ | $550_{\pm 50.5}$ | $640_{\pm 30.7}$ |

Table 2: Comparison of proposed approaches against baselines on **train** environments. Results for Fourrooms and Pointmass are averaged across 20 mazes configurations.

| Environment (Train) | Method | | | | | |
|---|---|---|---|---|---|---|
| | Random | Vanilla-FB | HILP | Lap | **Belief-FB** | **Rotation-FB** |
| Randomized-Fourrooms | $0.18_{\pm 0.02}$ | $0.25_{\pm 0.02}$ | $0.4_{\pm 0.02}$ | $0.2_{\pm 0.1}$ | $0.7_{\pm 0.02}$ | $0.85_{\pm 0.02}$ |
| Randomized-Pointmass | $0.0_{\pm 0.05}$ | $0.2_{\pm 0.2}$ | $0.45_{\pm 0.1}$ | $0.15_{\pm 0.15}$ | $0.76_{\pm 0.18}$ | $0.88_{\pm 0.2}$ |
| Ant-Wind | $-190_{\pm 250}$ | $390_{\pm 120}$ | $410_{\pm 90}$ | $340_{\pm 150}$ | $680_{\pm 80}$ | $740_{\pm 70}$ |

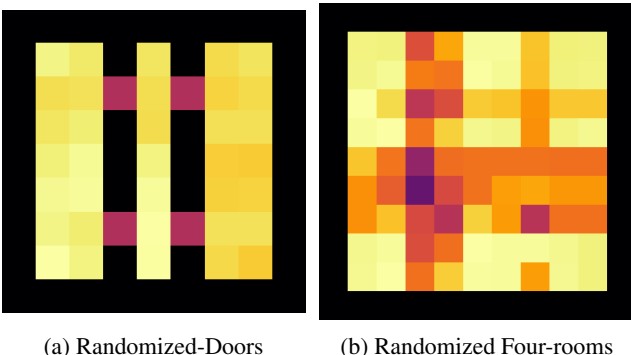

(a) Randomized-Doors     (b) Randomized Four-rooms

Figure 11: State occupancy measures visualizations for collected datasets for discrete-based environments.

## C.1 Dataset Generation

For Randomized Four-Rooms, we produce four training datasets with the following parameters:

| # Transitions | # layouts | # episodes per layout | episode length |
|---|---|---|---|
| 1000000 | 10 | 1000 | 100 |
| 1000000 | 20 | 500 | 100 |
| 1000000 | 30 | 250 | 100 |
| 1000000 | 50 | 150 | 100 |

Table 3: Details for Randomized Four-Rooms datasets

**Randomized Four-Rooms.** For experiments on Randomized Four-Rooms during dataset collection we generate randomly grid layout, ensuring that each room is interconnected by exactly one door. For evalution we fix agent start position to $(1, 1)$ with the goal of reaching 3 other goals, specified at other rooms. Each episode terminates after $100$ steps. The evaluation protocol is averaged success rate across 3 across 20 environments.

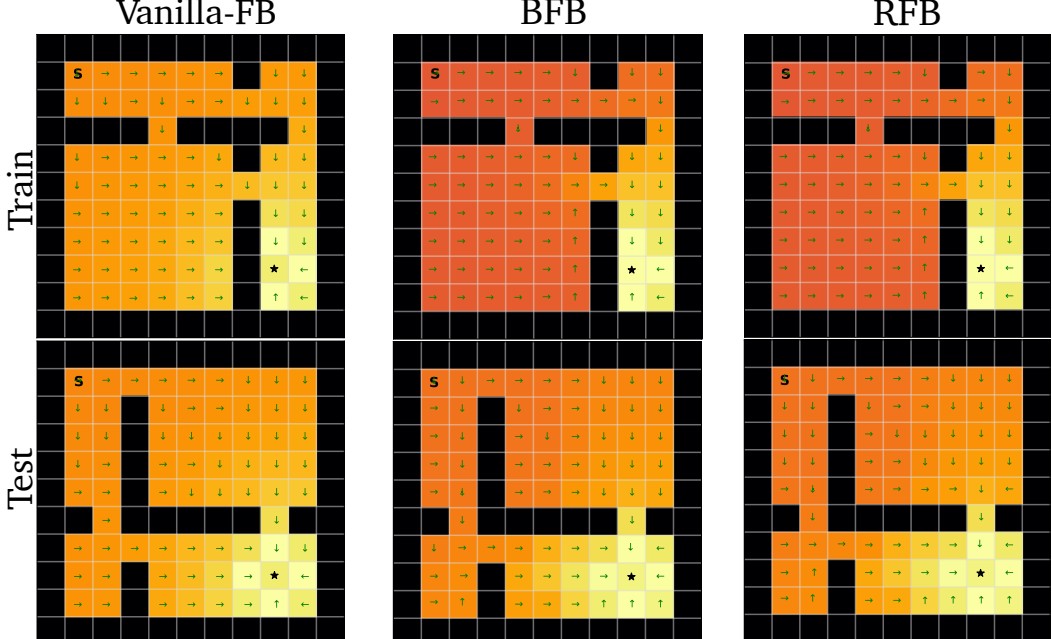

Figure 12: **Q-function and deterministic policy visualizations (Equation 4)** on Randomized Four-Rooms environment. Vanilla-FB ignores grid structure and resulting policy moves through obstacles. BFB and RFB do not have such issue.

**AntWind.** For AntWind we first collect trajectories by varying wind direction $d$ and training an expert-like SAC agent. After training, we collected evaluation trajectories from trained agent. This ensures that all directions are covered and explicitly sets dynamics context. As said in Experiments section, we train on 16 environments with wind directions corresponding to first 3 quadrants of circle, leaving other 4 (last quadrant) for hold out.

## D  Proofs

### D.1  Theorem 1

Preserving notation from Section 3.1, we provide a full proof of the Theorem 1. Let $\{M_{\pi_i}\}$ be a collection of successor measure of the optimal policies $\{\pi_i\}_{i=1}^k$ for $k$ distinct CMDPs. Given a reference measure $\rho$ on $\mathcal{S} \times \mathcal{A}$ let worst case regret be defined as

$$\epsilon_k := \inf_{F,B} \max_{i \leq i \leq k} ||M_{\pi_i} - F(\cdot, \cdot, z_i)^T B(\cdot)||_{L_\rho^2} \tag{11}$$

**Theorem** (Regret-bound for Multiple Dynamics). *Then, for any bounded $||r_\infty|| \leq R$ and any CMDP whose state-action distribution $\rho_{test}$ (assuming absolute continuity, i.e., $d\rho_{test}/\rho$ is bounded), the policy extracted from $F, B$ for that CMDP satisfies:*

$$\mathbb{E}_{(s,a) \sim \rho_{test}}[Q^{\pi^*}(s,a) - Q^{\pi_{z_i}}(s,a)] \leq \frac{2\gamma\epsilon_k ||r||_\infty}{(1-\gamma)^2}$$

*Since $\epsilon_{k+1} \geq \epsilon_k$ (monotonicity) the worst case regret per any CMDP at test time increases as more environments are included during training.*

**Lemma 1.** *Theorems 8-9 from Touati and Ollivier [38] prove this inequality for single instance of MDP, showing that if FB approximation error in $L^2(\rho)$ is at most $\epsilon$ then pointwise value gap is bounded by:*

$$(Q_r^* - Q_r^{\pi_{z_i}}) \leq \frac{\gamma}{1-\gamma}(P_{\pi^*} - P_{\pi_z})(I - \gamma P_{\pi^*})^{-1}E(z)r \tag{12}$$

*with $E(z)$ being a point-wise error matrix over state-actions as $E(z) = M^{\pi_z}(s,a,s') - F(s,a,z)^T B(s,a)$. Since*

$$||(I - \gamma P)^{-1}||_\infty \leq \frac{1}{1-\gamma} \tag{13}$$

923 *results in coefficient $2\gamma/(1-\gamma)^2$ in Equation 1.*

924 *Proof.* Define a transition kernel $P_i$ of CMDP at index $i$ and $M_{\pi_i}$ its successor measure. Let
925 $E_i = M_{\pi_i} - F(s,a,z_i)^T B(\cdot) = M_{\pi_i} - \hat{M}_i$. Then, using $Q^* = (I - \gamma P_{\pi^*})^{-1} r$ value gap
926 decomposes as

$$Q^* - Q^{\pi_{z_i}} = \gamma(I - \gamma P_{\pi^*})^{-1}(P_{\pi_*} - P_{\pi_{z_i}})(I - \gamma P_{\pi_{z_i}})^{-1} r \tag{14}$$

927 Since each of the resolvent factors (denote them as $E_i$)are at most $1/(1-\gamma)$ in $L^\infty$, then from
928 triangle inequality:

$$||Q^* - Q^{\pi_{z_i}}||_\infty \leq \frac{2\gamma}{(1-\gamma)^2} ||E_i||_{L^2_\rho} ||r||_\infty \tag{15}$$

929 From Assumption 1 on absolute continuity,

$$\mathbb{E}_{(s,a)\sim\rho_{\text{test}}}\{Q^* - Q^{\pi_{z_i}}\} \leq ||Q^* - Q^{\pi_{z_i}}||_\infty \tag{16}$$

930 Substituting this into Equation 15, gives desired inequality bound in Theorem 1. $\qquad\square$

## D.2 Theorem 2

932 Section 3.3 introduced a new sampling procedure of $z_{\text{FB}}$, which improves upon usual uniform
933 sampling. This procedure can also be studied more formally.

934 Given an $L$ possible contexual representations $h$ of the environments coming from $f_{\text{dyn}}$, define a
935 *cone* around each of the context axes $\{h^1, h^2 \ldots h^L\} \in \mathbb{S}^{d-1}$, with the angle between any two latent
936 vectors $\theta_{\max}$ set

$$C_j = \{z_{\text{FB}} \in \mathbb{S}^{d-1} | \langle z_{\text{FB}}, h^j \rangle \geq \cos\theta_{\max}\} \tag{17}$$

937 Corresponding policy task vectors are defined for each cone $z_{\text{FB}}^i \in C_{c(i)}$, with $c(i) \in \{1, \ldots L\}$
938 being a classification function, mapping index $i$ to one of the predifined context axes. For functions
939 $F, B$ define per environment error as:

$$\mathcal{E}_i(F, B) := ||M^{\pi_i} - F(\cdot, \cdot, z_{FB}^i)^T B(\cdot)||_{L^2(\rho)} \tag{18}$$

940 With following optimization tasks:

$$\epsilon_k := \inf_{F,B} \max_{1 \leq i \leq k} \mathcal{E}_i(F, B), \quad \epsilon_j := \inf_{F,B} \max_{i \in \mathcal{S}_j} \mathcal{E}_i(F, B) \tag{19}$$

941 with $\mathcal{S}_j = \{i | c(i) = j\}$ being a set of task vectors ($z_{\text{FB}}$) indices that fall into the $j$-th cone of the
942 latent space partition.

943 **Theorem** (Regret-bound under latent space partitioning). *Under assumptions above, the Gram matrix*
944 *of the directions $\{z_{FB}\}_{i=1}^k$ is block diagonal w.r.t. partition $\{S_j\}$ and*

$$\epsilon_k = \max_{j \leq L} \epsilon_j, \quad \epsilon_k \leq \epsilon_{k_{max}} \tag{20}$$

945 *with $k_{max} := \max_j |S_j|$ being the size of a largest cone block.*

946 In order to prove this theorem, assume that collection of contexual embeddings $\{h_j\}_{i=1}^L$ obtained
947 from $L$ environments are almost orthogonal.

948 *Proof.* Define a $k \times k$ Gram matrix as $G = \langle z_{\text{FB}}^i, z_{\text{FB}}^j \rangle$ with $i, j$ corresponding to cone partition.
949 Because cones, corresponding to different contexual embeddings $h$, are disjoint and lie in a span$\{h_i\}$,
950 the resulting Gram matrix is block diagonal $G = \text{diag}(G^{(1)}, G^{(2)}, .., G^L)$. For a fixed rank $d$ of $F, B$,
951 the worst case approximation error is

$$\epsilon_k(F, B) = \max_{1 \leq i \leq k} ||M_{\pi_i} - \hat{M}_{\pi_i}||_{L^2(\rho)} = \max_{j \leq L} \max_{i \in S_j} ||M_{\pi_i} - \hat{M}_{\pi_i}||_{L^2(\rho)} \tag{21}$$

952 Since matrix $G$ is block-diagonal, optimization of $F, B$ decouples over blocks of $G$. Namely,
953 minimizer on the full set is obtained by minimizing each block separately, hence:

$$\epsilon_k = \inf_{F,B} \epsilon_k(F, B) = \max_{j \leq L} \epsilon_j \tag{22}$$

954 By taking $k_{\max} = \max_j |S_j|$ and $\epsilon_k \leq \epsilon_{k_{\max}}$ for each block, we obtain desired inequality. $\qquad\square$

955 Notably, such orthogonal cone partitioning eliminates interference. Once each cone has its own
956 slice of the latent space, adding more cones does not enlarge the worst-case error bound, and with
957 representation capacity of F and B being $d \geq k_{\max}$ the FB model can reach zero approximation error
958 in principle.

Table 4: **Hyperparameters for FB** The additional hyperparameters for Belief-FB and Rotation-FB are highlighted in

| Hyperparameter | Value |
|---|---|
| Latent dimension $d$ | 150 (100 for discrete) |
| $F$ / $\psi$ dimensions | (1024, 1024) |
| $B$ / $\varphi$ dimensions | (256, 256, 256) |
| Preprocessor dimensions | (1024, 1024) |
| Std. deviation for policy smoothing $\sigma$ | 0.2 |
| Truncation level for policy smoothing | 0.3 |
| Learning steps | 1,000,000 |
| Batch size | 1024 |
| Optimiser | Adam |
| Learning rate | 0.0001 |
| Learning rate of $f_{\text{dyn}}$ | 0.0001 |
| Discount $\gamma$ | 0.99 |
| Activations (unless otherwise stated) | GeLU |
| Target network Polyak smoothing coefficient | 0.05 |
| $z$-inference labels | 10,000 |
| $z$ mixing ratio | 0.5 |
| $\kappa$ | 50, 100 for Pointmass |
| Contexual representation $h$ dimension | 150 (100 for discrete) |
| Next state predictor $g_{\text{pred}}$ | (256, 256, 256) |

# E  Implementation Details

## E.1  Forward-Backward Representations

### E.1.1  GPUs

We run each experiment on 4 Nvidia 4090.

### E.1.2  Architecture

The forward-backward architecture described below mostly follows the implementation by [39]. All other additional hyperparameters are reported in Table 4.

**Forward Representation** $F(s, a, z)$. The input to the forward representation $F$ is always preprocessed. State-action pairs $(s, a)$ and state-task pairs $(s, z)$ have their own preprocessors which are feedforward MLPs that embed their inputs into a 512-dimensional space. These embeddings are concatenated and passed through a third feedforward MLP $F$ which outputs a $d$-dimensional embedding vector. Note: the forward representation $F$ is identical to $\psi$ used by USF so their implementations are identical (see Table 4).

**Backward Representation** $B(s)$. The backward representation $B$ is a feedforward MLP that takes a state as input and outputs a $d$-dimensional embedding vector.

**Actor** $\pi(s, z)$. Like the forward representation, the inputs to the policy network are similarly preprocessed. State-action pairs $(s, a)$ and state-task pairs $(s, z)$ have their own preprocessors which are feedforward MLPs that embed their inputs into a 512-dimensional space. These embeddings are concatenated and passed through a third feedforward MLP which outputs a $a$-dimensional vector, where $a$ is the action-space dimensionality. A `Tanh` activation is used on the last layer to normalise their scale. Note the actors used by FB and USFs are identical (see Table 4).

**Misc.** Layer normalisation and `Tanh` activations are used in the first layer of all MLPs to standardise the inputs as recommended in original paper for both discrete and continuous becnhmarks.

## E.2  Task Sampling Distribution $\mathcal{Z}$

**Vanilla-FB.** FB representations require a method for sampling the task vector $z$ at each learning step. [39] employ a mix of two methods, which we replicate:

1. Uniform sampling of $z$ on the hypersphere surface of radius $\sqrt{d}$ around the origin of $\mathbb{R}^d$,

2. Biased sampling of $z$ by passing states $s \sim \mathcal{D}$ through the backward representation $z = B(s)$. This also yields vectors on the hypersphere surface due to the $L2$ normalization described above, but the distribution is non-uniform.

We sample $z$ 50:50 from these methods at each learning step as in original work by [38].

**Rotation-FB.** After transformer $f_{\text{dyn}}$ pretraining stage, RFB at each gradient step chooses task-conditioning vector $z_{\text{FB}}$ based on **i)** context representation $h$ acting as axes coming from $f_{\text{dyn}}$ and **ii)** drawing task encoding vectors $z_{\text{FB}}$ around this axes. We also perform normalization as in Vanilla-FB by projecting resulting vector on a surface of hypersphere of radius $\sqrt{d}$.

**Stage ii)** is implemented as drawing samples as $z_{\text{FB}} \sim \text{vMF}(\mu = h, \kappa)$. In order to remove high computational costs, we implement this sampling procedure through Householder reflection around context axes, by first drawing $z$ from one of the basis vectors (*e.g.*, north pole) and then performing rotation. This is depicted Pseudocode section Section 1:

### E.3 Pseudocode

---
**Algorithm 1** Belief-FB Training
---
1: **Input**: offline diverse dataset $\mathcal{D}$ consisting of transitions based on hidden configuration variable $c_i$
2: Initialize transformer encoder $f_{\text{dyn}_\theta}$, $F_\eta$, $B_\omega$, number of gradient steps for transformer pre-training $K$, context length $T$, Polyak coefficient, $\beta$, batch size $B$ learning rates $\lambda_f, \lambda_F, \lambda_B$
3: **while** update steps $< K$ **do**
4:     sample batch of $B$ trajectories of length $T$ $\{(s_{i,t}, a_{i,t}, s_{i,t+1})\}_{i=1,\dots B, t=1,\dots,T} \sim \mathcal{D}$
5:     $(\boldsymbol{\mu}_i; \log \boldsymbol{\sigma}_i), = f_{\text{dyn}_\theta}\big(\{s_{i,t}, a_{i,t}, s_{i,t+1}\}_{t=1}^M\big), i = 1, \dots, B,$
6:     $\boldsymbol{z_i} = \boldsymbol{\mu}_i + \boldsymbol{\epsilon}_i \odot \exp\big(\log \boldsymbol{\sigma}_i\big),$
7:     $\mathbf{Z}_{i,t} = \boldsymbol{z}_{\text{dyn}_i}, \ t = 1, \dots, T$    # Representation $z_{\text{dyn}}$ is shared across each sequence
8:     $\hat{s}_{i,t+1} = g_{\text{pred}}(s_{i,t}, a_{i,t}, \mathbf{Z}_{i,t})$      $t = 1, \dots, T, \ i = 1, \dots, B$
9:     $\mathcal{L}_{\text{context}} = \frac{1}{B\,T} \sum_{i=1}^B \sum_{t=1}^T \big\|\hat{s}_{i,t+1} - s_{i,t+1}\big\|_2^2$
10:    $\theta_{f_{\text{dyn}}} \leftarrow \theta_{f_{\text{dyn}}} - \lambda_f \nabla_\theta \mathcal{L}_{\text{context}}(\theta)$
11: **end while**
12: **while** not converged **do**
13:    $\eta_F \leftarrow \eta_F - \lambda_F \nabla_{\eta_F} J_{(F,B)}(\eta_F)$    # FB training, Equation 10
14:    $\omega_B \leftarrow \omega_B - \lambda_B \nabla_{\omega_B} J_{(F,B)}(\omega_B)$
15: **end while**

---

---
**Algorithm 2** Sampling $z_{\text{FB}}$ for RFB
---
**Input:** $B$ (batch size), $d$ (latent dimension), anchor matrix $\mathbf{H} \in \mathbb{R}^{B \times d}$, $\kappa$ (concentration)
**Output:** $\mathbf{Z} \in \mathbb{R}^{B \times d}$
1: **Normalize anchors:** $\mathbf{u}_i \leftarrow \mathbf{H}_i / (\|\mathbf{H}_i\|_2 + \varepsilon)$             $\triangleright$ for $i = 1, \dots, B$
2: $\mathbf{S} \leftarrow \text{VMF\_SAMPLE\_NORTHPOLE}(B, d, \kappa)$          $\triangleright$ draw $B$ VMF samples
3: **for** $i \leftarrow 1$ **to** $B$ **do**
4:    $\mathbf{R}_i \leftarrow \text{HOUSEHOLDER\_ROTATION}(\mathbf{u}_i)$
5:    $\mathbf{z}_i \leftarrow \mathbf{R}_i \mathbf{S}_i$
6: **end for**
7: $\mathbf{Z} \leftarrow \text{PROJECT\_TO\_SPHERE}\big(\{\mathbf{z}_i\}_{i=1}^B\big)$
8: **return** $\mathbf{Z}$

---

