# OpenReview forum: "Zero-Shot Adaptation of Behavioral Foundation Models to Unseen Dynamics"
_NeurIPS.cc/2025/Conference — Submitted to NeurIPS 2025_

### Official Review · Reviewer_cQsh · 2025-06-30

**Clarity:** 4
**Significance:** 3
**Originality:** 3
**Rating:** 5
**Confidence:** 4

**Summary:**

This paper proposes methods to enable zero-shot adaptation of Behavioral Foundation Models (BFMs) to environments with unseen transition dynamics. The authors identify that a BFM with vanilla Forward–Backward representation (FB) fails to generalize across different dynamics because it conflates experiences from multiple environments, causing “interference” among learned policies. To overcome this, the paper introduces (1) Belief-FB (BFB), which conditions the FB model on a latent belief state of the environment’s dynamics using a transformer encoder, and (2) Rotation-FB (RFB), which further partitions the latent policy space by dynamics by sampling policy vectors from a von Mises-Fisher distribution aligned with the inferred context. Theoretical results show that vanilla FB’s performance degrades as more environments with different dynamics are included, whereas the proposed RFB can remove this dependence. Empirical evaluations on gridworld-like navigation, a pointmass maze, and a MuJoCo Ant with varying wind demonstrate that BFB and RFB achieve substantially higher zero-shot returns than baselines for both seen and unseen dynamics.

**Questions:**

1. What future developments do you think could improve the scalability of your methods, so it can handle more complex environments or much longer trajectories?
2. With insights from this paper, do you think it’s possible to train a single BFM that generalizes not only across different dynamics of the same type of environment, but also across different environment types (e.g. both Randomized-Doors and Randomized-Pointmass)?

**Ethical Concerns:**

["NO or VERY MINOR ethics concerns only"]

**Final Justification:**

The authors have adequately addressed my questions in the rebuttal, and I will maintain my current score (Accept).

**Limitations:**

yes

**Paper Formatting Concerns:**

No paper formatting concerns

**Quality:**

3

**Strengths And Weaknesses:**

**Strengths**
1. The paper is well-structured and clearly written. It includes a comprehensive literature review and clear background info, and the proposed methods are novel and well-motivated.
2. Convincing experiments have been performed on both discrete and continuous environments with error bars reported. BFB and RFB show large performance improvements over state-of-the-art baselines on multiple tasks for both seen and unseen dynamics. Visualization of latent space and ablation studies on data diversity and context length further confirm the paper’s claims about disentanglement and context capture.
3. The paper provides regret bounds analysis that shows the worst case regret grows with the number of environments (Theorem 1) and suggests partitioning the latent space into non-overlapping clusters can eliminate that growth depending on the number of training environments k (Theorem 2).

**Weaknesses**
1. The context encoder $f_{dyn}$ is trained to infer dynamics-related hidden variables from visitation data (although it can be reward-free and generated by random policy) of the test environment. In the zero-shot evaluation, the agent needs to be given a “small batch of transitions” from the new context. In real-world application, obtaining such visitation data from the new environment could still be a challenge.
2. The scalability of current context encoder $f_{dyn}$ design is limited. Accommodating a larger number of distinct dynamics would likely require (1) increasing the dimensionality of the latent space h to separate all modes, (2) collecting a larger and more diverse offline dataset to cover every new context, and (3) growing the transformer’s capacity to process longer or higher-dimensional input. Together, these raise both sample complexity and compute/memory costs, potentially making real-time inference on resource-constrained platforms infeasible .
3. Empirical runtime is not reported. Such discussion would further clarify the approach’s computing resource requirement and efficiency.

---

> ### Author Rebuttal · Authors · 2025-07-30
>
> We are pleased that you found our approach and results promising, and we appreciate your careful assessment and valuable suggestions. Below, we provide detailed responses to your concerns and questions.
>
> > Q: Exploration at test-time could be a challenge for harder tasks
>
> We fully agree that relying on random exploration at test time is a limitation of our approach, and that a more nuanced exploration strategy should be employed (we kindly refer to the response for Reviewer b5Lr). Such strategies would better handle more complex environments and support more accurate identification of the underlying dynamics. However, in this work, we focus on a more fundamental issue: how to effectively train the Forward-Backward (FB) representation on data from multiple CMDPs. Notably, existing BFMs struggle even in simple cases involving dynamics variation (see our illustrative example in Section 3.1) and result in suboptimal policies. **Changing exploration strategy preserves all properties of BFB/RFB**
>
> > Q: What future developments could improve scalability of RFB/BFB?
>
> **One promising future direction for scaling is continual adaptation. In realistic scenarios, the environment may undergo multiple changes in its underlying state (e.g., the appearance of new objects), requiring the agent to continually adapt its behavior.** In such settings, identifying a suitable behavior policy $\pi_z$ that can account for these changes is essential for achieving optimal performance across tasks. Currently, we are not aware of any existing approaches that successfully scale BFMs to the continual learning regime.
>
> An interesting avenue for future work is to study the geometry of the learned behaviors under such non-stationarity, extending our analysis in Figures 3 and 4 and to understand how the internal representations evolve. For instance, it would be valuable to examine how  anchor embeddings $h$ shift instantaneously in response to changes in dynamics from a Bayesian perspective. This could shed light on stability of the learned behavioral basis and guarantees for optimal policies.
>
> **Another promising direction for scaling our approach is language grounding.** Rather than relying solely on sequences of trajectories to learn representations of dynamics, the agent could incorporate language-based descriptions during training. This would allow the model to generalize better at test time by leveraging high-level semantic information, e.g prior knowledge about object locations or environment structure, which is provided through natural language. Such grounding could significantly improve the agent ability to infer dynamics and adapt its behavior in unseen or partially observed environments.
>
> > Q: Computational requirements and scaling
>
> We should point out that our choice of permutation-invariant $f_{\text{dyn}}$ encoder (transformer) was made due to their impressive ability to encode sequences. However, other architectural choices are possible (e.g recent State Space Models) [1]. **Importantly, our core observations and contributions regarding policy space partitioning and context representation are agnostic to the specific architecture used for encoding dynamics.**
>
> [1] Gu et al. Efficiently Modeling Long Sequences with Structured State Spaces
>
> > Q: Possibility to perform cross-environment adaptation
>
> This is a very interesting and valuable question, and we thank the reviewer for proposing such an insightful experimental direction.
>
> **We believe that this form of generalization is indeed possible and represents a natural continuation of our work.** However, it would likely require an additional level of abstraction over the policy space $\pi_z$. Intuitively, learning BFMs that generalize across different environment types would involve two key steps: first, identifying the environment type, such as determining the index of a latent "sphere" of policies appropriate for that environment (analogous to the multi-variant case of Figure 4); and second, selecting a suitable behavior within that specific policy subspace, as was done in our approach.
>
> That said, the learning procedure becomes significantly more challenging, since estimating the successor measure $M^{\pi_z}$ would now require accurately capturing visitation occupancies across all policies $\pi_z$, all environments, and all dynamics. To address this, it may be necessary to introduce additional conditioning mechanisms, designed carefully to preserve theoretical guarantees, such as ensuring the possibility of extending Theorem 2 to this broader setting.
>
> > Q: Runtime evaluation
>
> Our entire training pipeline is implemented in JAX, which significantly accelerates training. Across all benchmarks, the total runtime, including both the transformer-based dynamics encoder and the Forward-Backward (FB) training, ranges from approximately 30 to 40 minutes on an NVIDIA RTX 4090 GPU, depending on the specific task.

---

> ### Comment · Reviewer_cQsh · 2025-08-02
>
> Thank you for your thoughtful response! I will maintain my current score (Accept) and increase the confidence score since the authors' rebuttal adequately addresses my concerns & questions.

---

### Official Review · Reviewer_RXmB · 2025-07-01

**Clarity:** 3
**Significance:** 3
**Originality:** 3
**Rating:** 5
**Confidence:** 1

**Summary:**

The paper investigates zero shot adaption of behavioral foundation models (BFMs) based on successor features. It has two parts: First they identify that the forward backward representation (FB) cannot distinguish between distinct dynamics leading to interference among latent directions that define policies. This can lead to problems for the model to adapt to changes in the dynamics of the task/environment. As a fix for this issues, they propose a transformer based belief estimator (BFB) and an enhancement thereof (RFB). Using different RL tasks, they demonstrate that their two approaches outperforms existing approaches but a substantial margin.

**Questions:**

I don't have any immediate questions apart from a brief discussion on the scalability (see weaknesses).

**Ethical Concerns:**

["NO or VERY MINOR ethics concerns only"]

**Final Justification:**

The authors have addressed my concerns and I kept the accept score.

**Limitations:**

Are discussed in section 5.

**Paper Formatting Concerns:**

No concerns

**Quality:**

3

**Strengths And Weaknesses:**

Strengths:
- The paper is generally well written, figures are high quality, proofs and other details are provided in the Appendix.
- The paper addresses a relevant problem.
- The paper demonstrates substantial improvements on the chosen tasks.
- The paper provides theoretical backup for their claims and observations.

Weaknesses:
- There could be a bit more discussion whether and how the method can scale to bigger problems and what the challenges would be. However, the authors do mention scaling in future work, albeit only in one sentence.

---

> ### Author Rebuttal · Authors · 2025-07-30
>
> We thank reviewer for the thorough and encouraging evaluation. Below we address main concerns regarding scalability.
> > Q: How scalable proposed approach to bigger problems and possible challenges.
>
> At the current state, scaling Behavior Foundation Models (BFMs) to more challenging domains, such as robotics, remains an active research direction. A key limitation is performance degradation due to structural discrepancies in dynamics between the deployment environment and the simulator used for training. In our paper, we analyze why BFMs fail even in simpler settings and propose a solution. **For example, training FB on diverse cross-domain/embodiment data, obtained from various sources (e.g Open-X [1] or DROID [2]) would result in an averaged future predictive policy, a phenomenon we observed in our paper even in much simpler cases.** While our approach is effective in a simplified scenarios (tested benchmarks), several critical questions remain regarding scalability:
>
> 1. **How can we effectively balance exploration with optimal policy inference in continual learning scenarios?** While our current framework operates under static environment dynamics (where environment properties remain fixed during evaluation), real-world applications demand adaptive strategies to handle dynamic conditions (e.g., unexpected obstacles or changing terrains). Extending BFB/RFB to accommodate such non-stationary environments represents a promising direction for future research.
>
> 2. **Another challenge is related to high coverage assumption (Assumption 1 in paper)**: FB presumes there is offline data whose state-action distribution covers most of the state space. Violating this assumption can lead to narrow policy representations $\pi_z$, resulting in suboptimal performance. Extending FB/BFB/RFB to cases when high-coverage assumption is relaxed is other direction for scaling.
>
> In conclusion, **we believe our paper takes a step toward developing truly generalizable agents: ones that not only exhibit diverse behaviors but also dynamically adapt their actions to their surroundings.** We will add discussion in the final version of paper

---

> > ### Comment · Reviewer_RXmB · 2025-08-02
> > **Thanks**
> >
> > Thank you for the response. I'll keep my (accept) score.

---

### Official Review · Reviewer_b5Lr · 2025-07-03

**Clarity:** 3
**Significance:** 3
**Originality:** 3
**Rating:** 5
**Confidence:** 3

**Summary:**

The paper proposes to improve a Forward-Backward (FB)-based behavioral foundation model for zero-shot adaptation to unseen dynamics. With both theoretical analysis and empirical observations, the paper poses the policy averaging problem of the previous FB approach on unseen dynamics, and proposes two solutions that infer a hidden context variable.

**Questions:**

* In the experiment with randomized four-rooms, why does the test return decrease as the context length increases after ~100? Does this also happen in the pointmass task when the context length is further increased?

* Would the performance become optimal if we give an oracle trajectory that contains all possible transitions instead of the random-exploration-context?

**Ethical Concerns:**

["NO or VERY MINOR ethics concerns only"]

**Final Justification:**

As the authors have addressed my concerns, I raise my score to Accept.

**Limitations:**

yes

**Quality:**

3

**Strengths And Weaknesses:**

The method is theoretically grounded and well-motivated. The conceptual experiments provide insightful analyses on the behavior of the proposed methods and baselines in the dynamics-changing scenarios.

The major weakness is the reliance on random exploration. This can be limited in complex environments where a random policy cannot effectively cover the possible transitions, or the required context length should be largely increased. The inefficiency of the transformer architecture in encoding long sequences further amplifies the limitation.

---

> ### Author Rebuttal · Authors · 2025-07-30
>
> We sincerely thank the reviewer for their insightful feedback and constructive critique. Below, we address each of the questions and concerns with detailed explanations, along with additional experiments.
>
> > Q: In the experiment with randomized four-rooms, why does the test return decrease as the context length increases after ~100? Does this also happen in the pointmass task when the context length is further increased?
>
> We thank reviewer for asking this question and would like to clarify this a bit further:
>
> In our experiments, we used a context length no greater than the maximum number of episode steps in the environment (for Randomized Four-Rooms, this is $T=100$; for Randomized PointMass, it is $T=250$). Such design is also aligned with literature, e.g [1]. So observed decline on context sizes (Figure 5) greater than episode length stems from our design experiments: in this case, $f_{\text{dyn}}$ during training can process trajectories from different layouts as a single sequence, leading to ambiguous representation of $h$, which should predict both dynamics simultaniously (i.e violating one-one mapping correspondence), and, as a result, degrading performance for both BFB and RFB.
>
> To disambiguate our results and show that increasing context does not lead to worse performance, we conducted additional experiments where $f_{\text{dyn}}$ accepts several episodes **from the same layout type**:
>
> For Randomized-PointMass $(\text{max episode length} \: T=250)$:
> | | Context=250 (Train) | Context=350 (Train) |
> |--------|---------------------|---------------------|
> | BFB    | 0.76 ± 0.15         | 0.77 ± 0.10         |
> | RFB    | 0.85 ± 0.20         | 0.86 ± 0.10         |
>
> | | Context=250 (Test) | Context=350 (Test) |
> |--------|--------------------|--------------------|
> | BFB    | 0.42 ± 0.02        | 0.42 ± 0.02        |
> | RFB    | 0.54 ± 0.05        | 0.57 ± 0.05        |
>
> Results above depict marginal improvements as context increases. For RFB, this aligns with Theorem 2: once context length suffices for identification of a single dynamics, extra transitions only add noise, potentially increasing $\epsilon_{k_{\text{max}}}$ in the error bound.
>
> In summary, a context length matching or slightly exceeding the episode length proves sufficient, as additional transitions beyond this point become redundant and may degrade the quality of the learned belief embedding. We thank the reviewer for pointing this inconsistency out. We will include the updated figures for such edge cases into the final version of the paper.
>
> [1] Ni et al., When Do Transformers Shine in RL?
> Decoupling Memory from Credit Assignment (NeurIPS 2023)
>
> >Q: Would the performance become optimal if we give an oracle trajectory that contains all possible transitions instead of the random-exploration-context?
>
> We thank the reviewer for their interesting question. Yes, providing oracle (or expert-like) trajectory during test-time, especially with dynamics identifying information, helps significantly in inferring the most suitable latent representation of dynamics $h$, leading to improved performance.
>
> Informally, random exploration may visit irrelevant states (i.e., those that do not reduce uncertainty) or fail to expose critical dynamics, resulting in a noisier $h$. However, while the oracle trajectory improves belief estimation (via conditioning on $h$), it cannot by itself close approximation gap $\epsilon_k$ inherited from finite rank approximation of the occupancy measure via FB. Thus, performance may approach but not necessarily reach optimality.
>
> To support claims above, we conducted following experiment: we compared random exploration policy and oracle policy (implemented as Q-learning) in Randomized-PointMass environment, for both train and test dynamics:
> | Train | Randomized | Expert(Q-learning) |
> | --- | --- | --- |
> | BFB | $0.76 \pm0.18$ |$ 0.78 \pm 0.1$ |
> | RFB | $0.86 \pm 0.2$ | $0.9\pm 0.1$ |
> |  |  |  |
>
>
> | Test| Randomized | Expert(Q-learning) |
> | --- | --- | --- |
> | BFB | $0.42 \pm 0.02$ | $0.48 \pm 0.03$ |
> | RFB | $0.57 \pm 0.05$ | $0.62 \pm 0.03$ |
> |  |  |  |
>
> Moreover, providing optimal trajectory can be viewed as a way to *extract the closest* imitation learning policy from trained behaviors across all dynamics via $a = \arg \max_a F(s, a, [z;h]]^T ([z;h])$, which draws connection to cross-domain imitation learning (in terms of dynamics mismatches).
>
> From the experiments above, we observe that both RFB and BFB benefit from trajectories that nearly fully describe the environment, as such trajectories provide sufficient information to infer $h$ and ensure generalization. However, enhancing BFB/RFB with test-time exploration strategies that balance oracle-like and random-like behavior is an interesting direction for future research.
>
> > Q: Relying on random exploration during testing could hinder the recovery of the dynamics representation.
>
> We fully agree that random exploration in highly complex environments may fail to discover crucial states needed to disambiguate dynamics identification. However, we emphasize that our work addresses a distinct bottleneck: existing behavioral foundation models (BFMs), particularly FB, tend to collapse when trained on offline data composed of mixed CMDPs. Consequently, **training BFMs on recently introduced large-scale robotics datasets (e.g., Open-X [1]) would yield an averaged policy, thus limiting their current applicability to unimodal datasets (in terms of dynamics mismatch)**. Our work demonstrates how BFB/RFB overcomes this collapse. Developing smarter test-time exploration strategies to streamline dynamics identification remains an important direction for future research.
>
> For context, one could utilize predictive models of the future (e.g., γ-models [2]) or implement planning over goal-conditioned landmarks [3], as done in prior work on successor features [3], without modifying the core principles of BFB/RFB. In our experiments, we found that random exploration was sufficient to identify and generalize across novel dynamics types. We will also update our paper with results obtained with Q-learning expert data from the previous answer.
>
> [1] Open X-Embodiment: Robotic Learning Datasets and RT-X Models
>
> [2] M. Janner et al., Gamma-Models: Generative Temporal Difference Learning for Infinite-Horizon Prediction (NeurIPS 2020)
>
> [3] Hoang et al., Successor Feature Landmarks for Long-Horizon
> Goal-Conditioned Reinforcement Learning (NeurIPS 2021)
>
> >Q: $f_{\text{dyn}}$ encoder complexity
>
> We respectfully emphasize that our primary contribution lies in identifying a crucial limitation of Behavioral Foundation Models (BFMs), particularly methods based on estimating successor measures (e.g., FB), which hinders their generalization across diverse dynamics mismatches. Our proposed solutions (RFB/BFB) employ transformers due to their sequence modeling capabilities and emergent properties like in-context learning [1, 2]. However, we do not advocate for a fixed choice of $f_{\text{dyn}}$ and in principle, other architectures with lower computational costs (e.g., SSMs, LSTMs, GRUs) could be equally viable.
>
> [1] Furuya et al. Transformers are Universal In-context Learners (ICLR 2025)
>
> [2] Wei et al. Emergent Abilities of Large Language Models (TMLR 2024)

---

> > ### Author Response · Authors · 2025-08-04
> >
> > As the discussion period draws to a close, we would greatly appreciate it if you could review our response and let us know if there are any remaining points you would like us to clarify. We hope we have addressed your concerns and are happy to continue the discussion.

---

> ### Comment · Reviewer_b5Lr · 2025-08-05
>
> Thank you for the thoughtful response. The authors have addressed my concerns, so I raise my score to Accept.

---

### Note · Authors · 2025-08-13

We thank the reviewers for their constructive feedback and the opportunity to summarize our responses. We believe we have addressed all the questions raised during the discussion period and provided comprehensive answers, with additional emphasis on potential future directions stemming from our work. Below we summarize our responses:

1) **Scalability**: Our method directly addresses the core bottleneck of BFMs, particularly their tendency to collapse toward averaged behaviors when trained on an offline data coming from multiple Contexual Markov Decision Processes (CMDPs). We tackle this issue both theoretically (Theorems 1 and 2) and empirically (Section 3.1 for a failure of original FB on a toy example and Section 4 for full experiments). By resolving this limitation, our approach enables BFMs to be trained on vast, unlabeled datasets with varying dynamics (e.g., Open-X) while still producing optimal policies for any downstream task at test time.

The primary computational bottleneck lies in the belief-encoder ($f_{\text{dyn}}$), which we implement as a permutation-invariant transformer. However, we emphasize that our core contributions for solving $\pi_z$ collapse problem via Belief-FB dynamics encoding and its extension via policy-space partitioning (RFB), are architecture-agnostic. While we use transformers in our experiments, alternative architectures (e.g., state-space models (SSMs), RNNs etc) could also be employed, potentially lowering computational burden

2) **Random exploration at test time**: We agree that random test-time exploration is ineffective for dynamics identification in much complex environments, and future work should develop smarter strategies like uncertainty-driven planning or language-conditioned prior to guide agents toward informative states. In the current work we proposed ways to solve collapse issue

3) **Future research directions**: Incorporating **language grounding** could enhance our approach by combining test-time trajectory sequences with high-level semantic descriptions, enabling better generalization and adaptation in unseen environments through natural language prior. This integration would leverage prior knowledge about object locations and environment structure to improve dynamic inference and behavior adaptation. Another extension is **continual adaptation** in dynamically changing environments, where optimal policy $\pi_z$ should be recomputed each time, which is much general case compared to our work

---

### Decision · Program_Chairs · 2025-09-17

**Decision:**

Reject

**Comment:**

This paper builds on the premise Behavioral Foundation Models (BFMs) to enable generalization to new, unseen transition dynamics. The authors show that standard FB-based methods fail to generalize to new dynamics because experiences from different environments are conflated, causing interference. To rectify this, the authors propose 2 methods - BFB and RFB, which do belief estiamtion with a transformer based architecture to condition the prediction. The authors then evaluate this empirically on gridworld and simulated navigation problems, showing generalizations to new dynamics.

All reviewers were strongly supportive of the paper and it has both empirical and theoretical merit. It is a valuable addition to the literature in this space!

===

As recently advised by legal counsel, the NeurIPS Foundation is unable to provide services, including the publication of academic articles, involving the technology sector of the Russian Federation’s economy under a sanction order laid out in Executive Order (E.O.) 14024.

Based upon a manual review of institutions, one or more of the authors listed on this paper submission has ties to organizations listed in E.O. 14024. As a result this paper has been identified as falling under this requirement and therefore must not be accepted under E.O. 14024.

This decision may be revisited if all authors on this paper can provide proof that their institutions are not listed under E.O. 14024 to the NeurIPS PC and legal teams before October 2, 2025. Final decisions will be communicated soon after October 2nd. Appeals may be directed to pc2025@neurips.cc.